# Body-Weight Fluctuations and the Association Between the Consumption of Protein-Rich Foods and the Incidence of Metabolic Syndrome Among Middle-Aged Women in Korea

**DOI:** 10.3390/healthcare13070709

**Published:** 2025-03-24

**Authors:** Hyejin Chun, Jung-Heun Ha, Jongchul Oh, Miae Doo

**Affiliations:** 1Department of Family Medicine, Ewha Womans University College of Medicine, Seoul 07804, Republic of Korea; fmewha@naver.com; 2Department of Food Science and Nutrition, Dankook University, Cheonan 31116, Republic of Korea; ha@dankook.acc.kr; 3Department of Mathematics, Kunsan National University, Jeonbuk 54150, Republic of Korea; ohjc@kunsan.ac.kr; 4Department of Food and Nutrition, Kunsan National University, Jeonbuk 54150, Republic of Korea

**Keywords:** body-weight fluctuation, dietary consumption, Korean genome and epidemiology study, metabolic syndrome, middle-aged women, protein-rich foods

## Abstract

Background/Objectives: Metabolic syndrome (MetS) is a growing global health concern, driven in part by increasing rates of overweight and obesity. In Korea, MetS incidence escalates particularly among middle-aged women, eventually surpassing that of men. While protein-rich diets have been associated with improved metabolic health, the impact of protein intake on body weight fluctuations (BWFs) and MetS risk has received limited attention, especially in Korean populations. Methods: Using data from the Korean Genome and Epidemiology Study (KoGES), this study examined whether a higher intake of protein-rich foods is linked to smaller BWF and lower MetS incidence in middle-aged Korean women. Dietary intake was assessed through validated questionnaires, and BWF was calculated based on repeated anthropometric measurements over a long-term follow-up. MetS was defined according to established clinical criteria. Results: Preliminary findings indicated that participants who consumed higher amounts of protein-rich foods, particularly animal-based proteins (e.g., fish, low-fat dairy), tended to exhibit smaller BWFs. Additionally, lower BWF was associated with a reduced risk of MetS, suggesting that stable weight regulation may play a protective role against metabolic dysfunction. Potential mechanisms include the preservation of lean mass, enhanced energy expenditure, and reduced carbohydrate intake when protein consumption is increased. These findings highlight the importance of dietary strategies that optimize protein intake to help minimize BWF and potentially lower MetS risk in middle-aged Korean women. Future research should investigate the specific sources and quality of protein and their long-term effects on metabolic health outcomes in diverse populations.

## 1. Introduction

Globally, the prevalence of metabolic syndrome (MetS) is on the rise, and as individuals increasingly experience weight gain—a major risk factor for developing MetS—this trend contributes to the expanding burden of the syndrome. MetS is characterized by a cluster of risk factors, including high blood pressure, elevated blood glucose, increased triglycerides, reduced high-density lipoprotein (HDL) cholesterol, and an enlarged waist circumference [1]. For example, a meta-analysis reported prevalence rates ranging from 12.5% to 31.4% in adults, noting that Asian and Hispanic groups in the United States experienced a faster increase compared to other populations [2,3]. In Korea, recent data from the Korea National Health and Nutrition Examination Survey (KNHANES, 2019–2021) indicate a MetS prevalence of 24.9% among adults aged 19 and older, with an overall lower rate in women (19.0%). However, among middle-aged women (40–60 s), prevalence escalates with age—from 14.2% to 38.4%—and eventually surpasses that in men after middle age, underscoring an urgent need for targeted prevention and management strategies [4,5].

People often strive to achieve an ideal body weight composition, and numerous weight loss trials are executed with this goal in mind. However, inconsistent lifestyle changes can inadvertently increase the risk of body weight fluctuations (BWF). Maintaining a healthy body weight is generally associated with a reduced risk of disease and an enhanced quality of life. Nonetheless, the relationship is complex and potentially bidirectional, encompassing both anabolic and catabolic responses; weight may change due to disease onset or treatment during a study, which can lead to discrepancies between baseline and follow-up measurements [6]. BWFs, which include both minor daily variations and longer-term weight cycling, serve as important indicators of health status. Research has linked significant BWFs with an increased risk of various chronic conditions, including cardiovascular disease, diabetes, hypertension, cancer, dementia, and overall mortality [7,8,9,10]. Notably, a 7-year follow-up study of a middle-aged French cohort identified weight variability as an independent risk factor for MetS [11].

Recent studies have highlighted the significant impact of protein intake on metabolic health. Diets rich in protein are consistently linked to reductions in fasting blood glucose, waist circumference, and improvements in lipid profiles—factors that are critical in managing metabolic syndrome (MetS) [12,13,14]. Moreover, protein-rich diets aid in weight loss and maintenance by enhancing satiety, thereby reducing overall caloric intake, and by preserving fat-free mass and resting energy expenditure during weight changes [15,16]. Despite these benefits, emerging evidence indicates that the source and quality of protein—whether plant- or animal-based—play a pivotal role in these positive outcomes. However, protein consumption in Korea has declined slightly since 1998, with current averages at 73.9 g for men and 53.7 g for women, which are relatively low compared to other populations [17]. These findings underscore the importance of nutritional strategies that optimize protein intake to mitigate risks associated with weight fluctuations and subsequent chronic conditions such as MetS.

This study aims to evaluate the relationship between protein-rich food consumption and body weight fluctuations (BWFs) among Korean women, and to determine whether these differences are significantly associated with MetS morbidity. Utilizing data from the Korean Genome and Epidemiology Study (KoGES), our research employs a comprehensive analytical approach to assess how variations in protein intake may influence BWF and, in turn, metabolic health outcomes. By integrating dietary assessments, anthropometric measurements, and longitudinal health data, the study seeks to provide a nuanced understanding of the interplay between protein consumption, weight stability, and MetS risk. The potential outcomes of this research may inform public health strategies, contribute to the refinement of dietary guidelines, and support the development of targeted interventions aimed at reducing the incidence of chronic metabolic diseases in populations with similar dietary patterns and health profiles.

## 2. Subjects and Methods

### 2.1. Data Source and Selection of Subjects

The Korean Genome and Epidemiology Study (KoGES) is an ongoing large population-based prospective cohort study initiated in 2001 to investigate chronic diseases and their risk factors [18]. The study recruited participants from the urban Ansan and rural Ansung areas, who completed health and lifestyle surveys, underwent biochemical examinations and health checkups, and were followed up every 2 years. Among the initial 10,030 participants, women *(n* = 5272) with inadequate or missing data were excluded as follows: those who did not participate in consecutive follow-ups (*n* = 1265), those with MetS at baseline (*n* = 681), those consuming <500 or >3500 kcal/day (*n* = 229), or those with missing data (*n* = 484). After these exclusions, a total of 2613 Korean women were included in this study. All participants provided written informed consent, and the study was approved by the Institutional Review Board of Kunsan National University (Ethical approval code 1040117-202304-HR-006-01).

### 2.2. Data Collection and MEASURES

Data on general characteristics, anthropometric and biochemical variables, and consumption of protein-rich foods were obtained from the KoGES data.

#### 2.2.1. General Characteristics

General characteristics—including age, residence area, educational level, household income, alcohol consumption, smoking status, physical activity, and disease diagnoses—were collected through structured interviews. The Ansan and Ansung cohorts were categorized by residential area, distinguishing between urban and rural regions. Educational level was classified as “≤middle school” or “≥high school”, while household income was dichotomized into “low” or “high” based on the median value. Current alcohol intake and smoking were defined as “yes” if the subject was presently consuming alcohol or smoking, and “no” if they had never engaged in these behaviors or had quit. Physical activity was quantified by considering the intensity and duration of various activities (including sedentary behavior, work-related activities, strenuous activities, and exercise), and was expressed in Metabolic Equivalent of Task (MET) units as follows [19].Total physical activity (MET) = ∑ [Physical activity intensity (MET) × time required (time)]

Disease diagnosis was defined as a self-reported physician diagnosis of hypertension, type 2 diabetes, cardiovascular disease, or hyperlipidemia.

#### 2.2.2. Anthropometric Biochemical Variables

Anthropometric measurements, including height, body weight, and waist circumference, were obtained using standardized protocols. Body mass index (BMI) was calculated as weight (kg) divided by the square of height (m^2^). Body weight fluctuation (BWF) was assessed based on within-individual changes over the follow-up period, using at least three consecutive weight measurements across nine follow-up surveys. BWF indices were determined using standard deviation (*SD)*, coefficient of variation (*CV*), average real variability (*ARV*), and variation independent of the mean (*VIM*). The formulas used were as follows:i.The standard deviation (*SD*) was calculated according to the formula below:$$SD=\sqrt{\frac{1}{N}\sum_{i=1}^{N}\left|Bw_{i}-\bar{Bw}\right|}$$
where *N* is the number of body-weight measurements and *BW* is the measured body weight.
ii.The coefficient of variation (*CV*) was defined as the ratio of the standard deviation to the mean of body weight multiplied by 100, and was calculated as *CV* = (*SD*/mean of body weight) × 100.iii.The average real variability (*ARV*) was calculated according to the following formula:

$$ARV=\frac{1}{N-1}\sum_{i=1}^{N-1}\left|{BW}_{i+1}-{BW}_{i}\right|\;$$
where *N* is the number of body-weight measurements and *BW* is the measured body weight

The *VIM* was calculated via the following formula.$$VIM=\frac{Standard\;deviation\;of\;body\;weight}{{Mean\;of\;body\;weight}^{\beta }}\times 100$$

*β* is the regression coefficient obtained by regressing the natural logarithm of *SD* on the natural logarithm of the mean body weight. Given concerns that *SD* and *ARV* are influenced by average body weight, and that *CV* may not fully eliminate this effect, *VIM* was used as the primary index for BWF in this study [20]. *VIM* values were categorized into “low body-weight fluctuation (LBWF)” and “high body-weight fluctuation (HBWF)” groups based on the median value.

Blood pressure was measured after a 5 min rest in the supine position, with the average of three readings taken at 30 s intervals used for analysis. Fasting glucose, triglycerides, total cholesterol, and HDL-cholesterol levels were measured using a Hitachi Automatic Analyzer 7600 (Hitachi, Tokyo, Japan). MetS was defined based on the modified NCEP/ATP III criteria and the waist circumference criteria for abdominal obesity recommended by the Korean Society for the Study of Obesity [21,22]. Specifically, MetS was defined as the presence of three or more of the following: (1) waist circumference ≥ 85 cm; (2) triglycerides ≥ 150 mg/dL or treatment for hypertriglyceridemia; (3) HDL cholesterol < 50 mg/dL or treatment for hypo-HDL-cholesterolemia; (4) blood pressure ≥ 130/85 mmHg or treatment for hypertension; and (5) fasting glucose ≥ 100 mg/dL or treatment for hyperglycemia.

#### 2.2.3. Dietary Assessment

Dietary intake was assessed using a semi-quantitative food frequency questionnaire (FFQ) developed and validated for the KoGES [23,24]. The FFQ, which includes 103 food items, evaluated the frequency and portion size of food consumed over the past year. The correlation coefficients between the FFQ and a 12-day dietary record for various nutrients were approximately 0.45, indicating acceptable validity [24]. Average macronutrient consumption was calculated by multiplying the frequency of each food item by its nutrient content per portion (based on the Korean Nutrient Database) and summing these values across all food items.

Based on an analysis using the 2017 KNHANES and the Korean Nutrient Database, the protein content per serving for major protein sources is as follows: meat and eggs (7.5–13.8 g), fish and shellfish (7.5–22.6 g), beans and seeds (7.2–7.7 g), and milk and dairy products (5.2–6.2 g). Protein-rich foods were therefore defined as “meat and eggs”, “fish and shellfish”, “legumes and nuts”, and “milk and dairy products”, with consumption divided by the median intake (see Table 1). To account for differences in energy intake, dietary macronutrient and protein-rich food consumption were expressed as grams per 1000 kcal.

### 2.3. Statistical Analysis

Categorical variables are presented as percentages, and continuous variables as means with standard errors (assuming a normal distribution). Baseline differences in general characteristics, MetS-related variables, and protein-rich food consumption according to BWF were assessed using t-tests or Fisher’s exact tests. Incidence of MetS was defined as meeting three or more of the diagnostic criteria during follow-up in participants free of MetS at baseline. Cox proportional hazard regression was used to calculate hazard ratios (HRs) and 95% confidence intervals (CIs) for the incidence of MetS, using high BWF as the reference group. Both crude and adjusted models were applied, with the adjusted model controlling for age, residence area, education level, household income, current alcohol consumption, current smoking, and physical activity. To evaluate the impact of protein-rich food consumption on the association between BWF and MetS incidence, protein-rich food intake was dichotomized based on the median, and Cox regression models were applied with covariate adjustment. Sensitivity analyses were performed excluding subjects who were postmenopausal at baseline. All statistical tests were two-sided, with a significance level set at *p* < 0.05. Data were analyzed using SPSS version 27.0 (IBM Corp., Armonk, NY, USA).

## 3. Results

The general characteristics and indices of body weight variability according to body weight fluctuations are presented in Table 2. The average ages for the low body weight fluctuation (LBWF) and high body weight fluctuation (HBWF) groups were 50.46 and 51.22 years, respectively. Compared with subjects in the HBWF group, those in the LBWF group had higher educational levels (39.1% vs. 34.5%, *p* = 0.017) and higher household incomes (51.8% vs. 45.1%, *p* = 0.001). Additionally, the proportion of subjects diagnosed with chronic diseases (including hypertension, diabetes, cardiovascular disease, and hyperlipidemia) was lower in the LBWF group (15.2%) than in the HBWF group (21.1%, *p* < 0.001). However, no significant differences were observed in residence area, current alcohol consumption, smoking status, or physical activity. All indices of body-weight fluctuation—including standard deviation (SD), coefficient of variation (CV), variability independent of the mean (VIM), and average real variability (ARV)—differed significantly between the LBWF and HBWF groups (*p* < 0.001 for all).

Table 3 and Figure 1 show the metabolic syndrome (MetS)-related variables at baseline and the incidence of MetS by body weight fluctuations during follow-up. The body mass index (BMI) of the LBWF group was significantly lower than that of the HBWF group (24.38 kg/m^2^ vs. 25.43 kg/m^2^, *p* < 0.001). Compared with the HBWF group, the LBWF group had significantly lower waist circumference (79.31 cm vs. 82.56 cm, *p* < 0.001), systolic blood pressure (117.74 mmHg vs. 119.97 mmHg, *p* < 0.001), diastolic blood pressure (77.29 mmHg vs. 78.91 mmHg, *p* < 0.001), and fasting glucose (81.02 mg/dL vs. 87.52 mg/dL, *p* < 0.001). A Cox proportional hazard model indicated that a low fluctuation in body weight was associated with a significantly decreased incidence of MetS; subjects in the LBWF group had a 17% lower risk of developing MetS than those in the HBWF group (hazard ratio [HR], 0.83; 95% confidence interval [CI], 0.75 to 0.92; *p* < 0.001, Figure 2).

Because the association between weight variability and the incidence of MetS in middle-aged women (aged 40 to 60 years) may be affected by menopausal status, a sensitivity analysis was performed using a Cox proportional hazard model after excluding subjects with postmenopausal status. In this analysis, the HR for MetS incidence in subjects with low body weight fluctuation was 0.66 (95% CI, 0.55 to 0.80; *p* < 0.001) compared to those with high body weight fluctuation (Appendix A).

Table 4 presents the consumption of protein-rich foods at baseline according to body weight fluctuations. There was no significant difference in total energy consumption between the groups. Because food and nutrient consumption generally increases with energy intake, the intake of each food group or macronutrient was expressed per 1000 kcal of energy consumed. All macronutrient consumptions differed by body weight fluctuation status; subjects in the LBWF group consumed significantly more protein and less carbohydrates and fat than those in the HBWF group (*p* = 0.016 for protein, and *p* < 0.001 for both carbohydrates and fat). Moreover, the total amount of protein-rich food consumed by the LBWF group was significantly greater than that consumed by the HBWF group (141.71 g vs. 125.91 g, *p* < 0.001). Among the protein-rich food groups, consumption of fish and shellfish (20.51 g vs. 19.11 g, *p* = 0.024) and milk and dairy products (69.14 g vs. 55.97 g, *p* < 0.001) was significantly higher in the LBWF group than in the HBWF group.

After stratifying subjects based on the median consumption of each protein-rich food, hazard ratios (HRs) and 95% confidence intervals (CIs) were calculated for the association between body weight fluctuations and MetS incidence, controlling for covariates such as age, residence area, education level, household income, current alcohol consumption, smoking status, and physical activity (Figure 2). Among subjects with high total protein-rich food consumption, those in the LBWF group had an 18% lower risk of developing MetS compared with those in the HBWF group (HR, 0.82; 95% CI, 0.70 to 0.95; *p* < 0.05). In contrast, there were no significant differences in MetS incidence according to body weight fluctuations among subjects with low total protein-rich food consumption. Furthermore, among participants with high fish and shellfish consumption, the multivariate-adjusted HR for MetS incidence was 0.83 (95% CI, 0.71–0.96; *p* < 0.05) for the LBWF group relative to the HBWF group, and among those with high milk and dairy product consumption, the HR was 0.82 (95% CI, 0.70–0.95; *p* < 0.01). No significant association between body weight fluctuations and MetS incidence was observed among individuals with low consumption of meat, eggs, poultry, fish, shellfish, or milk and dairy products. Additionally, MetS incidence did not differ according to body weight fluctuations regardless of legume or nut consumption.

## 4. Discussion

This study identified differences in the consumption of protein-rich foods according to body weight fluctuations and suggested the possibility that protein-rich foods might modulate the incidence of metabolic syndrome according to body-weight variability, which is a novel strategy to prevent and manage metabolic syndrome in middle-aged Korean women. These meaningful results suggest that the incidence of metabolic syndrome is greater in the elderly population in Korea than in the middle-aged population, especially in women, compared with men. Its importance is also emphasized when considering that Korean women have a relatively low intake of protein-rich foods compared with other races or countries [9].

In general, maintaining a healthy weight throughout life by losing weight in overweight or obese people and gaining weight in underweight people is recommended. However, intentional or unintentional weight loss (or, for some, weight gain) can frequently lead to weight gain (or, for others, weight loss). These changes in body weight refer to body-weight fluctuations. Many studies have suggested that weight variability is associated with the risk of developing MetS, a finding that is consistent with our study [11,25,26]. On the one hand, a prospective cohort study in Koreans confirmed that although weight fluctuation over 16 years did not increase the risk of developing MetS, it was significantly associated with the risk of abdominal obesity, one of the diagnostic criteria for MetS [27]. The BWF is related to a complicated change of body-fat composition. In a study by Jacquet et al. where a mathematical model was constructed for body cycling, it was demonstrated that the loss of lean massis greater than loss of fat mass during weight loss and that fat mass is recovered more than lean mass during weight regain [28]. A study in animal models induced by weight cycling suggested that repeated weight cycling events increase body fat accumulation, thereby worsening metabolic dysfunction [29]. Additionally, the important meaning of body weight change over a certain period can be explained by a loss of physiological homeostasis, which can cause fluctuations in the body and ultimately increase the risk of negative health and chronic diseases [30,31].

Fortunately, many studies have shown that altering lifestyle, such as diet modulation, can help alleviate weight fluctuations. The high consumption of dietary protein may help reduce weight without affecting the preservation of lean body mass, regardless of physical activity [14,32], and result in the maintenance of stable body weight (or low BWF). Our findings suggest that high consumption of protein-rich foods is significantly associated with low BWF during long-term follow-up. A plausible explanation for these results is that consuming more protein-rich foods minimizes body composition changes (i.e., maintains muscle mass) despite frequent weight fluctuations, ultimately resulting in subjects with a lower BWF. Other plausible explanations include a systematic review and meta-analysis by Kim et al., which suggested that high-protein diets alter total energy expenditure, including by modulating the thermic effect, increasing resting energy expenditure, and promoting fat oxidation in overweight and obese subjects [32]. Animal protein, which contains high levels of essential amino acids, is usually considered to be more efficient at supporting muscle mass growth and recovery and maintaining muscle mass [33]. However, it is recommended to selectively supply protein foods because the intake of fish, milk and dairy products (low-fat foods) among animal-based protein and legumes and nuts among plant-based proteins lowers the risk of cardiovascular disease and is inversely correlated with hypertension, obesity, and insulin resistance [34]. In this study, the relationship with body weight was confirmed by dividing it into protein sources; animal foods, including meat, eggs, poultry, fish, shellfish, milk, and dairy products; and plant foods, including beans and nuts. In this study, subjects who consumed more fish, shellfish, milk or dairy products presented smaller fluctuations in body weight. These results suggest that protein sources such as fish, shellfish, or milk and dairy products may have the potential to maintain lean mass and result in a lower BWF.

In this study, we found that the association between BWF and the incidence of MetS was modulated by protein-rich food consumption. Although studies have recently reported an association between weight variability and MetS in terms of the consumption of dietary carbohydrates and refined grains, our results are the first to address the consumption of protein-rich food groups [26]. A low weight fluctuation had a protective effect on the risk of developing MetS among subjects who consumed more total protein food groups and animal-source food groups. The reasons for these results are unclear. The reason for these results is unclear, but protein consumption is considered to be due to individual differences in body composition changes and related metabolic changes resulting from these changes in weight during long-term follow-up. That is, consuming protein-rich foods, especially animal protein-rich foods, could reduce weight fluctuations, but because body composition and metabolic changes vary from person to person, protein-rich food intake does not have a positive effect on the risk of MetS.

Although this is the first study to report a novel perspective on the long-term weight change and protein-rich food-induced reduction in the risk of MetS in Korean women, which increases rapidly with age, several limitations should be acknowledged. First, although long-term weight changes were measured objectively and consistently throughout the follow-up period, there are issues related to weight changes due to hormonal changes, such as menopause, which is common in middle-aged women. In this context, it should be noted that while weight loss through lifestyle changes that result in long-term weight changes is related to a decreased risk of MetS, unintended weight loss is related to several health problems. Second, although this study selected several food groups as protein-rich foods, selection errors might have occurred across different food groups. Third, since this study is observational, there is no clear cause-and-effect relationship, and caution is needed in interpreting the findings. Finally, selection bias may be present due to the exclusion of subjects who did not meet the criteria for weight fluctuation, dietary intake, and metabolic syndrome variables during follow-up. Therefore, further research involving various age groups is needed to validate the findings of this study.

## Figures and Tables

**Figure 1 healthcare-13-00709-f001:**
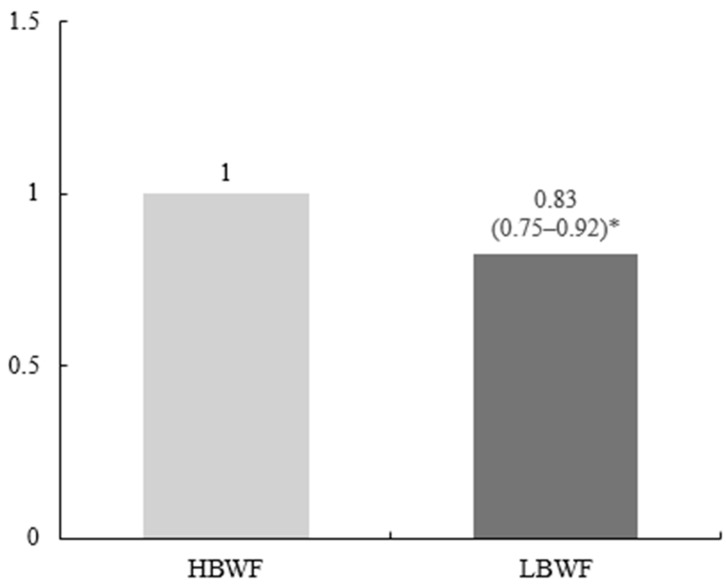
**Incidence of metabolic syndrome by body weight fluctuations.** The hazard ratio (95% confidence interval) and *p* value of the LBWF group were calculated using the HBWF group as the reference. * *p* value < 0.001. LBWF, low body weight fluctuation; HBWF, high body weight fluctuation.

**Figure 2 healthcare-13-00709-f002:**
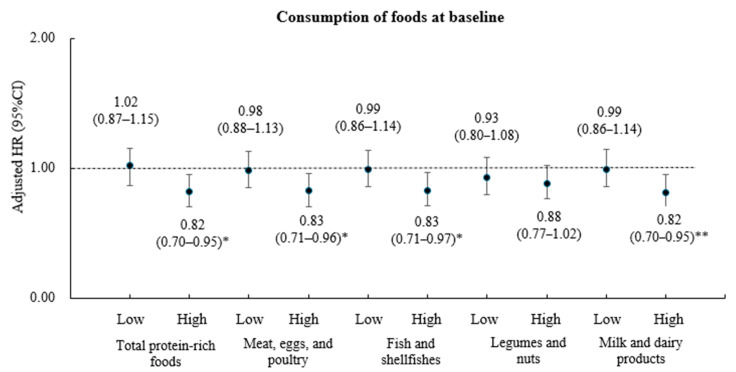
**Incidence of metabolic syndrome by protein rich foods and body weight fluctuations.** The hazard ratio (95% confidence interval) and *p* value of the LBWF group were calculated using the HBWF group as a reference after adjustment for age, residence area, education level, household income, current alcohol consumption, current smoking, and physical activity. * *p* value < 0.05; ** *p* value < 0.001. LBWF, low body weight fluctuation; HBWF, high body weight fluctuation.

**Table 1 healthcare-13-00709-t001:** Food sources of protein-rich foods in the Korean Genome and Epidemiology Study.

Food Group	Food List
**Animal-based food**	
Meat and eggs	Pork belly, grilled/stir-fried/bulgogi/donburi, steamed pork (bossam, jangjorim, pork trotters), processed meat (ham, sausage), by-products, steak, beef, dog meat, fried chicken, chicken backsuk, samgyetang, chicken doritang, soup (seolleongtang, gomtang, galbi tang, crucible tang, etc.), soup (beef soup, yukgaejang, etc.) Eggs/quail eggs
Fish and shellfish	Raw fish, mackerel, Pacific saury, Spanish mackerel, Hair tail, eel, yellow croaker, sea bream, flat fish, pollack, anchovy, cuttlefish, octopus, clam, oyster, crab, and shrimp, Squid/dried squid/octopus, canned tuna, jjigal (squid, spear roe, pollock roe, shrimp, anchovy, clam, etc.), clam/bonefish (including soup, stew, grilled, knife noodles, radish paste, etc.), fish cake/crab meat
Milk and dairy	Butter/margarine, milk, yogurt/yoplait, ice cream, cheese, cream in tea
**Plant-based food**	
Legumes and Nuts	Soybean/soybean Jaban, soup and stew with soybean paste/cheongguk, tofu, muk, soy milk, peanut/almond/pine nut

**Table 2 healthcare-13-00709-t002:** General characteristics and indices at baseline according to body weight fluctuations.

Variables	LBWF (n = 1297)	HBWF (n = 1316)	*p* Value
Age, years	50.46 ± 0.22	51.22 ± 0.23	0.018
Residence area, Ansan	54.4	50.6	0.060
Educational level, ≥high school	39.1	34.5	0.017
Household income, high	51.8	45.1	0.001
Current alcohol consumption, no	28.0	26.9	0.540
Current smoking, no	98.8	98.3	0.415
Physical activity, METS	9566.42 ± 167.55	9726.75 ± 167.38	0.498
Diagnosis of diseases, yes *	15.2	21.1	<0.001
Indices of body-weight variability			
Standard deviation	1.40 ± 0.01	3.00 ± 0.03	<0.001
Coefficient of variation	2.46 ± 0.02	5.09 ± 0.05	<0.001
Variation independent of the mean	0.79 ± 0.01	1.68 ± 0.02	<0.001
Average real variability	1.35 ± 0.01	2.28 ± 0.03	<0.001

The data are presented as the means ± SEs or %. *p* values were obtained via *t* tests or Fisher tests. * Diagnosis of diseases was defined as being diagnosed with one of the following diseases: hypertension, diabetes, cardiovascular disease, or hyperlipidemia. LBWF, low body weight fluctuation; HBWF, high body weight fluctuation.

**Table 3 healthcare-13-00709-t003:** Metabolic syndrome-related variables at baseline according to body weight fluctuations.

Variables	LBWF (n = 1297)	HBWF (n = 1316)	*p* Value
Body mass index, kg/m^2^	24.23 ± 0.08	25.43 ± 0.08	<0.001
Waist circumference, cm	79.31 ± 0.22	82.56 ± 0.17	<0.001
Systolic blood pressure, mmHg	117.74 ± 0.45	119.97 ± 0.44	<0.001
Diastolic blood pressure, mmHg	77.29 ± 0.29	78.91 ± 0.29	<0.001
Fasting glucose, mg/dL	87.52 ± 0.49	91.02 ± 0.49	<0.001
Triglycerides, mg/dL	132.99 ± 2.23	129.68 ± 2.21	0.294
Total cholesterol, mg/dL	197.56 ± 0.95	196.24 ± 0.94	0.322
HDL cholesterol, mg/dL	51.05 ± 0.32	51.10 ± 0.32	0.912

The data represent the means ± SEs. *p* values were obtained via *t* tests.

**Table 4 healthcare-13-00709-t004:** Consumption of dietary macronutrients and protein-rich foods at baseline by body weight fluctuation.

Variables	LBWF (n = 1297)	HBWF (n = 1316)	*p-*Value
Dietary macronutrient consumption			
Energy (kcal)	1824.34 ± 509.96	1822.35 ± 506.94	0.920
Carbohydrate (g)	179.01 ± 0.47	181.19 ± 0.45	0.001
Protein (g)	33.57 ± 0.16	33.02 ± 0.0.16	0.016
Fat (g)	15.60 ± 0.16	14.70 ± 0.16	<0.001
Total protein-rich food consumption			
Total protein-rich foods (g)	141.71 ± 2.44	125.91 ± 2.16	<0.001
Meat, eggs, and poultry (g)	30.27 ± 0.61	29.04 ± 0.62	0.156
Fish and shellfish (g)	20.57 ± 0.48	19.11 ± 0.43	0.024
Legumes and nuts (g)	21.74 ± 0.63	21.79 ± 0.64	0.952
Milk and dairy products (g)	69.14 ± 2.10	55.97 ± 1.81	<0.001

The data represent the means ± SE. *p*-values were obtained using *t*-test. The dietary macronutrients and protein rich foods are expressed as grams per 1000 kcal. LBWF, low body weight fluctuation; HBWF, high body weight fluctuation.

## Data Availability

The data used in this study are available from the National Institute of Health for Korea, CODA (Clinical & Omics Data Archive). Available online: https://coda.nih.go.kr/frt/index.do (accessed on 15 November 2023).

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
