# Peer review of "Body-Weight Fluctuations and the Association Between the Consumption of Protein-Rich Foods and the Incidence of Metabolic Syndrome Among Middle-Aged Women in Korea"

_healthcare, 2025, doi:10.3390/healthcare13070709_

Round 1
Reviewer 1 Report
Comments and Suggestions for Authors
The authors have undertaken an interesting topic and made a commendable effort to evaluate the risk of metabolic syndrome incidents among Korean middle-aged women in connection with body weight fluctuations and protein-rich food consumption. However, the presentation and discussion of the results leave room for improvement. In particular, several areas of concern require attention:
1. The Material and Methods section should be more concise (see comments in the manuscript).
2. The discussion of results should be clearer and avoid replicating tables as text.
3. Only the intake of protein-rich products was assessed, and there was no analysis of the total protein intake between the studied groups. Although there are significant differences in the protein-rich food intake between the groups, these are mainly due to differences in the consumption of milk and milk products. The results do not indicate whether consumption was higher for milk, fermented milk products or cheese. There are also significant differences (slightly) in fish consumption, but there is no explanation of the specific products and their protein content. If protein intake had been assessed directly, authors would be able to assess its impact on the risk of metabolic syndrome.
4. There are contradictory statements in several sections of the manuscript.
Given these concerns, I cannot recommend the manuscript for submission in its current state. However, with careful revisions and addressing the points raised, the manuscript has the potential to be a valuable contribution to the scientific community.

Author Response
The authors have undertaken an interesting topic and made a commendable effort to evaluate the risk of metabolic syndrome incidents among Korean middle-aged women in connection with body weight fluctuations and protein-rich food consumption. However, the presentation and discussion of the results leave room for improvement. In particular, several areas of concern require attention:
We thank the reviewer for careful reading and description about our manuscript with the valuable comments. We worked to the best of our abilities to revise the issues reviewer pointed out.
- The Material and Methods section should be more concise (see comments in the manuscript).
We really appreciate the constructive and very helpful comments. As your comment, “Subjects and Methods” section was revised, as follow; line 105-196
- The discussion of results should be clearer and avoid replicating tables as text.
The discussion section was modified to a different expression without clearly replicating tables.
- Only the intake of protein-rich products was assessed, and there was no analysis of the total protein intake between the studied groups. Although there are significant differences in the protein-rich food intake between the groups, these are mainly due to differences in the consumption of milk and milk products. The results do not indicate whether consumption was higher for milk, fermented milk products or cheese. There are also significant differences (slightly) in fish consumption, but there is no explanation of the specific products and their protein content. If protein intake had been assessed directly, authors would be able to assess its impact on the risk of metabolic syndrome.
We thank the reviewer for careful reading and description about our manuscript with valuable comments.
According to your comments, I have presented the total protein intake between the study groups, as follow; line 247-279.
“All macronutrient consumptions differed by body weight fluctuation status; subjects in the LBWF group consumed significantly more protein and less carbohydrates and fat than those in the HBWF group (p = 0.016 for protein, and p < 0.001 for both carbohydrates and fat).”
However unfortunately, the KoGES data used in our manuscript did not provide the nutrient content for each food group, so the protein content in each protein-rich product could not be provided.
We agree with your point that the difference in total protein rich foods intake between groups is due to the variation in milk and dairy product intake. However, even when milk and dairy products were excluded from total protein rich foods, there was a significant difference between groups. Also, the protein contained in fish was presented, as follow, line169-172.
“Based on an analysis using the 2017 KNHANES and the Korean Nutrient Database, the protein content per serving for major protein sources is as follows: meat and eggs (7.5–13.8 g), fish and shellfish (7.5–22.6 g), beans and seeds (7.2–7.7 g), and milk and dairy products (5.2–6.2 g)”
- There are contradictory statements in several sections of the manuscript.
We have thoroughly revised the sections of the manuscript that raised your concerns.
Given these concerns, I cannot recommend the manuscript for submission in its current state. However, with careful revisions and addressing the points raised, the manuscript has the potential to be a valuable contribution to the scientific community.
We have also made general revisions to the content pointed out in the PDF file.

Reviewer 2 Report
Comments and Suggestions for Authors
ABSTRACT:
"Chronic diseases incident" is grammatically incorrect. It should be "incidence of chronic diseases."
Abstract contains a lot of sentences that are grammatically incorrect, awkwardly structured and unclear.
The introduction should more clearly establish why the study is important and how protein intake and body weight fluctuation relate to MetS.
The results should present key takeaways first before diving into numerical details, making it easier for readers to grasp the study's significance.
INTRODUCTION:
Several grammatical and typographical errors make the text difficult to read.
The prevalence statistics are presented in a fragmented manner, making it harder to follow. It would be clearer to summarize these findings before providing numerical details.
The definition of MetS at the beginning should be more structured. The list of MetS components should follow a consistent format.
The paragraph discussing causality and biases (lines 48–52) is unclear. The discussion of time-based bias needs better wording to ensure clarity.
The discussion on protein intake and gender differences is somewhat disorganized. It jumps from general information about protein intake to gender-specific findings without a clear transition.
The definition of body weight fluctuation (BWF) is vague.
The claim that protein-enhanced diets reduce body weight in men but not in women (lines 66–67) should be supported with a more nuanced explanation or a stronger transition.
The statement about protein intake trends from 1998 to 2023 (lines 68–70) is not well integrated into the argument. How does this relate directly to BWF or MetS?
The introduction lacks a clear hypothesis or research objective at the end. The final paragraph attempts to provide one, but it is not well-structured.
The paragraph on lifestyle habits (lines 41–43) is somewhat misplaced. While lifestyle factors are important, they should be introduced in a more structured way rather than appearing as a standalone statement.
METHODS
There are also numerous grammatical errors, typos, and awkward phrasings throughout this section.
There is lack of clear study design description; The study design (longitudinal, observational, etc.) is not explicitly stated. It would be helpful to specify whether this is a prospective or retrospective cohort study. The exclusion criteria are mentioned vaguely but should be explicitly outlined.
The description of body-weight fluctuation (BWF) measurement is somewhat technical and difficult to follow. It would be clearer if broken down into: what BWF represents, why VIM was chosen, and how it was categorized into "low" and "high" fluctuation groups.
The sentence "The MetS status was assessed based on the NCEP/ATP III guidelines [20], excluding the abdominal obesity criterion, if three or more of the remaining diagnostic criteria were satisfied." is unclear.
The phrase "excluding the abdominal obesity criterion" contradicts the next line, which lists "Waist circumference ≥85 cm according to the Korean Society for the Study of Obesity" as a criterion. The diagnostic criteria should be presented in a clearer, bullet-point format:
The "Data Collection and Measures" section should be divided into:
Demographic and health-related variables
Anthropometric and biochemical measurements
Dietary assessment
The "Assessment of dietary intake" section is vague and lacks detail on: how portion sizes were measured, whether nutrient intake was validated, how missing data were handled.
RESULTS
It is unclear how many participants were excluded due to missing data. If a significant portion was excluded, it might introduce selection bias.
The semi-quantitative food frequency questionnaire (FFQ) is prone to recall bias and measurement errors.
Additionally, protein-rich food groups were categorized based on median consumption, which may not reflect meaningful dietary differences.
While the study adjusts for several variables (age, income, smoking, etc.), it does not account for factors such as stress, hormonal changes, or medication use, which may influence both weight fluctuations and metabolic syndrome risk.
The study relies heavily on p-values to determine significance but does not report effect sizes, which would help interpret the real-world impact of findings.
It is possible that individuals with metabolic syndrome have greater body-weight fluctuations rather than weight fluctuations leading to metabolic syndrome. The study does not account for this bidirectional relationship.
The study does not discuss whether sensitivity analyses were conducted to test the robustness of findings (e.g., excluding participants with preexisting conditions).
DISCUSSION:
While the study suggests that low BWF is associated with lower MetS risk, it cannot establish causality due to its observational nature.
The study claims that "protein-rich foods might modulate the risk of metabolic syndrome incidents according to weight variability," but the data only show an association, not causation. The language used suggests a stronger causal relationship than the study can support.
It also states that this is a "novel strategy" for managing metabolic syndrome without sufficiently acknowledging previous research on protein intake and weight management.
The study acknowledges individual differences in metabolism and body composition changes but does not explore how these differences might impact the findings.
The study reports that fish, shellfish, milk, and dairy products were associated with lower weight fluctuation, yet legumes and nuts did not show a similar effect. The discussion does not address why plant-based proteins might not contribute in the same way, leaving an important gap in the interpretation.
It is also unclear why certain animal-based proteins (e.g., meat and eggs) were not highlighted despite being included in the analysis.
The study acknowledges some limitations, such as menopause-related weight changes and potential selection errors in protein-rich food groups. However, it does not sufficiently discuss:
The reliability of dietary assessments (e.g., self-reported food intake in FFQs).
Reverse causation (i.e., whether individuals with stable weight patterns already have healthier dietary habits).
Unmeasured confounders (e.g., stress, exercise, or other dietary factors influencing weight fluctuations).
The study's inability to control for long-term adherence to dietary patterns—did participants consistently eat the same amount of protein over time?
While the study suggests that consuming protein-rich foods may help reduce weight fluctuations, it does not provide clear guidelines on what amounts or types of protein should be consumed.
It also fails to acknowledge potential risks of high protein intake, such as kidney strain or associations with other metabolic disorders.
The study does not provide clear recommendations on how findings can be applied in clinical or public health settings.
Comments on the Quality of English Language
Paper contains a lot of sentences that are grammatically incorrect, awkwardly structured and unclear.
Author Response
We thank the reviewer for careful reading and description about our manuscript with the valuable comments. We worked to the best of our abilities to revise the issues reviewer pointed out.
ABSTRACT:
"Chronic diseases incident" is grammatically incorrect. It should be "incidence of chronic diseases."
Abstract contains a lot of sentences that are grammatically incorrect, awkwardly structured and unclear.
The introduction should more clearly establish why the study is important and how protein intake and body weight fluctuation relate to MetS.
The results should present key takeaways first before diving into numerical details, making it easier for readers to grasp the study's significance.
All the contents pointed out in the Abstract section have been revised as follows, line 15-33.
“ Background/Objectives: Metabolic syndrome (MetS) is a growing global health concern, driven in part by increasing rates of overweight and obesity. In Korea, MetS incidence escalates particularly among middle-aged women, eventually surpassing that of men. While protein-rich diets have been associated with improved metabolic health, the impact of protein intake on body weight fluctuations (BWF) and MetS risk has received limited attention, especially in Korean populations. Methods: Using data from the Korean Genome and Epidemiology Study (KoGES), this study examined whether higher intake of protein-rich foods is linked to smaller BWF and lower MetS incidence in middle-aged Korean women (40s-60s). Dietary intake was assessed through validated questionnaires, and BWF was calculated based on repeated anthropometric measurements over a long-term follow-up. MetS was defined according to established clinical criteria. Results: Preliminary findings indicated that participants who consumed higher amounts of protein-rich foods, particularly animal-based proteins (e.g., fish, low-fat dairy), tended to exhibit smaller BWF. Additionally, lower BWF was associated with a reduced risk of MetS, suggesting that stable weight regulation may play a protective role against metabolic dysfunction. Potential mechanisms include the preservation of lean mass, enhanced energy expenditure, and reduced carbohydrate intake when protein consumption is increased. Conclusion: These findings highlight the importance of dietary strategies that optimize protein intake to help minimize BWF and potentially lower MetS risk in middle-aged Korean women. Future research should investigate the specific sources and quality of protein and their long-term effects on metabolic health outcomes in diverse populations.”
INTRODUCTION:
Several grammatical and typographical errors make the text difficult to read.
The prevalence statistics are presented in a fragmented manner, making it harder to follow. It would be clearer to summarize these findings before providing numerical details.
It has been modified based on the comment as follows, line 38-40, 51.
“Globally, the prevalence of metabolic syndrome (MetS) is on the rise, and as individuals increasingly experience weight gain—a major risk factor for developing MetS—this trend contributes to the expanding burden of the syndrome. “
“In Korea, recent data from the Korea National Health and Nutrition Examination Survey (KNHANES, 2019–2021) indicate a MetS prevalence of 24.9% among adults aged 19 and older, with an overall lower rate in women (19.0%). However, among middle-aged women (40s–60s), prevalence escalates with age—from 14.2% to 38.4%—and eventually surpasses that in men after middle age, underscoring an urgent need for targeted prevention and management strategies [4, 5].”
The definition of MetS at the beginning should be more structured. The list of MetS components should follow a consistent format.
It has been modified based on the comment as follows, line 40-42.
“MetS is characterized by a cluster of risk factors, including high blood pressure, elevated blood glucose, increased triglycerides, reduced high-density lipoprotein (HDL) cholesterol, and an enlarged waist circumference [1]”
The paragraph discussing causality and biases (lines 48–52) is unclear. The discussion of time-based bias needs better wording to ensure clarity.
It has been modified based on the comment as follows, line 56-59.
“Nonetheless, the relationship is complex and potentially bidirectional, encompassing both anabolic and catabolic responses; weight may change due to disease onset or treatment during a study, which can lead to discrepancies between baseline and follow-up measurements [6].”
The discussion on protein intake and gender differences is somewhat disorganized. It jumps from general information about protein intake to gender-specific findings without a clear transition..
It has been modified based on the comment as follows, line 68-77, -59.
“Moreover, protein-rich diets aid in weight loss and maintenance by enhancing satiety, thereby reducing overall caloric intake, and by preserving fat-free mass and resting energy expenditure during weight changes [15, 16]. Despite these benefits, emerging evidence indicates that the source and quality of protein—whether plant- or animal-based—play a pivotal role in these positive outcomes. However, protein consumption in Korea has declined slightly since 1998, with current averages at 73.9 g for men and 53.7 g for women, which are relatively low compared to other populations [17]. These findings underscore the importance of nutritional strategies that optimize protein intake to mitigate risks associated with weight fluctuations and subsequent chronic conditions such as MetS.”
The definition of body weight fluctuation (BWF) is vague.
It has been modified based on the comment as follows, line 52-54, and line 59-60.
People often strive to achieve an ideal body weight composition, and numerous weight loss trials are executed with this goal in mind. However, inconsistent lifestyle changes can inadvertently increase the risk of body weight fluctuations (BWF).
“BWF, which include both minor daily variations and longer-term weight cycling~”
The claim that protein-enhanced diets reduce body weight in men but not in women (lines 66–67) should be supported with a more nuanced explanation or a stronger transition.
We have been removed the pointed-out sections as they differed from the research background and objectives of this study.
The statement about protein intake trends from 1998 to 2023 (lines 68–70) is not well integrated into the argument. How does this relate directly to BWF or MetS?
We have made comprehensive revisions to the paragraph based on the feedback as follows, line 65-77
“Recent studies have highlighted the significant impact of protein intake on metabolic health. Diets rich in protein are consistently linked to reductions in fasting blood glucose, waist circumference, and improvements in lipid profiles—factors that are critical in managing metabolic syndrome (MetS) [12–14]. Moreover, protein-rich diets aid in weight loss and maintenance by enhancing satiety, thereby reducing overall caloric intake, and by preserving fat-free mass and resting energy expenditure during weight changes [15, 16]. Despite these benefits, emerging evidence indicates that the source and quality of protein—whether plant- or animal-based—play a pivotal role in these positive outcomes. However, protein consumption in Korea has declined slightly since 1998, with current averages at 73.9 g for men and 53.7 g for women, which are relatively low compared to other populations [17]. These findings underscore the importance of nutritional strategies that optimize protein intake to mitigate risks associated with weight fluctuations and subsequent chronic conditions such as MetS.”
The introduction lacks a clear hypothesis or research objective at the end. The final paragraph attempts to provide one, but it is not well-structured.
It has been modified based on the comment as follows, line 78-89.
“This study aims to evaluate the relationship between protein-rich food consumption and body weight fluctuations (BWF) among Korean women, and to determine whether these differences are significantly associated with MetS morbidity. Utilizing data from the Korean Genome and Epidemiology Study (KoGES), our research employs a comprehensive analytical approach to assess how variations in protein intake may influence BWF and, in turn, metabolic health outcomes. By integrating dietary assessments, anthropometric measurements, and longitudinal health data, the study seeks to provide a nuanced understanding of the interplay between protein consumption, weight stability, and MetS risk. The potential outcomes of this research may inform public health strategies, contribute to the refinement of dietary guidelines, and support the development of targeted interventions aimed at reducing the incidence of chronic metabolic diseases in populations with similar dietary patterns and health profiles.”
The paragraph on lifestyle habits (lines 41–43) is somewhat misplaced. While lifestyle factors are important, they should be introduced in a more structured way rather than appearing as a standalone statement.
It has been modified based on the comment as follows, line 52-54.
“People often strive to achieve an ideal body weight composition, and numerous weight loss trials are executed with this goal in mind. However, inconsistent lifestyle changes can inadvertently increase the risk of body weight fluctuations (BWF).”
METHODS
There are also numerous grammatical errors, typos, and awkward phrasings throughout this section.
There is lack of clear study design description; The study design (longitudinal, observational, etc.) is not explicitly stated. It would be helpful to specify whether this is a prospective or retrospective cohort study.
It has been modified based on the comment as follows line 92-93.
“The KoGES is an ongoing large population-based prospective cohort study~”
The exclusion criteria are mentioned vaguely but should be explicitly outlined.
It has been modified based on the comment as follows, line 96-101.
“Among the initial 10,030 participants recruited into the KoGES, women subjects (n=5,272) with inadequate and missing data were excluded in this study as follows; those who did not participate at consecutive follow-up (n=1,265), had metabolic syndrome at the start of the study (n=681), consumed of < 500 or >3500kcal/day (n=229), or did not offer missing data (n=484). After the exclusions, a total of 2,613 Korean women who participated in the KoGES were used for this study.”
The description of body-weight fluctuation (BWF) measurement is somewhat technical and difficult to follow. It would be clearer if broken down into: what BWF represents, why VIM was chosen, and how it was categorized into "low" and "high" fluctuation groups.
It has been modified based on the comment as follows, line126-148.
“BWF indices were determined using standard deviation (SD), coefficient of variation (CV), average real variability (ARV), and variation independent of the mean (VIM). The formulas used were;
- The standard deviation (SD)was calculated according to the formula:
,
where N is the number of body-weight measurements and BW is the measured body weight.
- The coefficient of variation (CV) was defined as the ratio of the standard deviation to the mean of body weight multiplied by 100, and was calculated as CV=(SD/mean of body weight)*100.
- The average real variability (ARV) was was calculated according to the formula;
,
where N is the number of body-weight measurements and BW is the measured body weight
- The VIM was calculated via the following formula.
,
β is the regression coefficient obtained by regressing the natural logarithm of SD on the natural logarithm of the mean body weight. Given concerns that SD and ARV are influenced by average body weight, and that CV may not fully eliminate this effect, VIM was used as the primary index for BWF in this study. VIM values were categorized into “low body-weight fluctuation (LBWF)” and “high body-weight fluctuation (HBWF)” groups based on the median value.”
The sentence "The MetS status was assessed based on the NCEP/ATP III guidelines [20], excluding the abdominal obesity criterion, if three or more of the remaining diagnostic criteria were satisfied." is unclear.
It has been modified based on the comment as follows, line152-154.
“The MetS status was assessed based modified NCEP/ATP III guidelines and the waist circumference criteria for abdominal obesity suggested by the Korean Society for the Study of Obesity [20, 21].
The phrase "excluding the abdominal obesity criterion" contradicts the next line, which lists "Waist circumference ≥85 cm according to the Korean Society for the Study of Obesity" as a criterion. The diagnostic criteria should be presented in a clearer, bullet-point format:
It has been modified based on the comment as follows, line154-158.
“Specifically, MetS was defined as the presence of three or more of the following: (1) waist circumference ≥85 cm; (2) triglycerides ≥150 mg/dL or treatment for hypertriglyceridemia; (3) HDL-cholesterol <50 mg/dL or treatment for hypo-HDL-cholesterolemia; (4) blood pressure ≥130/85 mmHg or treatment for hypertension; and (5) fasting glucose ≥100 mg/dL or treatment for hyperglycemia.
The "Data Collection and Measures" section should be divided into:
Demographic and health-related variables
Anthropometric and biochemical measurements
Dietary assessment
It has been modified based on the comment thorught .
The "Assessment of dietary intake" section is vague and lacks detail on: how portion sizes were measured, whether nutrient intake was validated, how missing data were handled.
It has been modified based on the comment as follows, line 161-176, and Table 1.
“Dietary intake was assessed using a semi-quantitative food frequency questionnaire (FFQ) developed and validated for the KoGES [23, 24]. The FFQ, which includes 103 food items, evaluated the frequency and portion size of food consumed over the past year. The correlation coefficients between the FFQ and a 12-day dietary record for various nutrients were approximately 0.45, indicating acceptable validity [24]. Average macronutrient consumption was calculated by multiplying the frequency of each food item by its nutrient content per portion (based on the Korean Nutrient Database) and summing these values across all food items.
Based on an analysis using the 2017 KNHANES and the Korean Nutrient Database, the protein content per serving for major protein sources is as follows: meat and eggs (7.5–13.8 g), fish and shellfish (7.5–22.6 g), beans and seeds (7.2–7.7 g), and milk and dairy products (5.2–6.2 g). Protein-rich foods were therefore defined as “meat and eggs,” “fish and shellfish,” “legumes and nuts,” and “milk and dairy products,” with consumption divided by the median intake (see Table 1). To account for differences in energy intake, dietary macronutrient and protein-rich food consumption were expressed as grams per 1,000 kcal.”
RESULTS
It is unclear how many participants were excluded due to missing data. If a significant portion was excluded, it might introduce selection bias.
As you pointed out, we agree that selection bias may exist. However, for the purposes of this study, it was necessary to exclude certain subjects during the selection process. This limitation has been addressed in the Discussion section as follows, line 359-362
“Finally, selection bias may be present due to the exclusion of subjects who did not meet the criteria for weight fluctuation, dietary intake, and metabolic syndrome variables during follow-up. Therefore, further research involving various age groups is needed to validate the findings of this study.”
The semi-quantitative food frequency questionnaire (FFQ) is prone to recall bias and measurement errors.
The FFQ used in our study has already been validated and is considered a reliable tool for assessing dietary intake in the Korean population as follows, line 161-165.
“Dietary intake was assessed using a semi-quantitative food frequency questionnaire (FFQ) developed and validated for the KoGES [23, 24]. The FFQ, which includes 103 food items, evaluated the frequency and portion size of food consumed over the past year. The correlation coefficients between the FFQ and a 12-day dietary record for various nutrients were approximately 0.45, indicating acceptable validity [24]. “
- Ref 23. Ahn, Y., Lee, J., Paik, H., Lee, H, Jo, Development of a semi-quantitative food frequency questionnaire based on dietary data from the Korea National Health and Nutrition Examination Survey. Nutr Sci. 2003, 6, 173–184.
- Ref 24. Ahn, Y., Kwon, E., Shim, J.E., Park, M.K., Joo, Y., Kimm, K., Park, C., & Kim, D.H. Validation and reproducibility of food frequency questionnaire for Korean genome epidemiologic study. Eur J Clin Nutr. 2007, 61, 35–41.
Additionally, protein-rich food groups were categorized based on median consumption, which may not reflect meaningful dietary differences.
As you pointed out, we acknowledge the limitation of not fully addressing meaningful dietary differences. This issue has been discussed in the Discussion section as follows, line 356-357.
“Second, although this study selected several food groups as protein-rich foods, selection errors might have occurred across different food groups.”
While the study adjusts for several variables (age, income, smoking, etc.), it does not account for factors such as stress, hormonal changes, or medication use, which may influence both weight fluctuations and metabolic syndrome risk.
We also consider the factors you pointed out to be important. Therefore, in this study, we adjusted for disease status (as medication use was based on a physician’s diagnosis). However, due to the lack of objective data on hormonal changes and stress levels, we were unable to adjust for these factors.
Nevertheless, we conducted a sensitivity analysis by classifying participants based on menopausal status to examine differences in metabolic syndrome incidence related to weight fluctuation as follow, line 193-194 and line229-233.
“Sensitivity analyses were performed excluding subjects who were postmenopausal at baseline.
“Because the association between weight variability and the incidence of MetS in middle-aged women (aged 40 to 60 years) may be affected by menopausal status, a sensitivity analysis was performed using a Cox proportional hazards model after excluding subjects with postmenopausal status. In this analysis, the HR for MetS incidence in subjects with low body weight fluctuation was 0.66 (95% CI, 0.55 to 0.80; p < 0.001) compared to those with high body weight fluctuation (Supplementary Figure 1).”
The study relies heavily on p-values to determine significance but does not report effect sizes, which would help interpret the real-world impact of findings.
As mentioned in the Discussion section, this study is an observational study, so caution is needed in interpreting the results as follow line 357-359. While we have suggested possible associations, we believe that the findings serve as fundamental data rather than directly providing practical guidance for real-world applications.
“Third, since this study is observational, there is no clear cause-and-effect relationship, and caution is needed in interpreting the findings.”
It is possible that individuals with metabolic syndrome have greater body-weight fluctuations rather than weight fluctuations leading to metabolic syndrome. The study does not account for this bidirectional relationship.
Like your comment, we agreed that this study does not provide cause-and-effect relationship. These points were presented in the discussion section as follows line 359-359.
“Third, since this study is observational, there is no clear cause-and-effect relationship, and caution is needed in interpreting the findings.”
The study does not discuss whether sensitivity analyses were conducted to test the robustness of findings (e.g., excluding participants with preexisting conditions).
According to your comments, we conducted a sensitivity analysis by classifying participants based on menopausal status to examine differences in metabolic syndrome incidence related to weight fluctuation as follow, line 193-194 and line229-233.
“Sensitivity analyses were performed excluding subjects who were postmenopausal at baseline.
“Because the association between weight variability and the incidence of MetS in middle-aged women (aged 40 to 60 years) may be affected by menopausal status, a sensitivity analysis was performed using a Cox proportional hazards model after excluding subjects with postmenopausal status. In this analysis, the HR for MetS incidence in subjects with low body weight fluctuation was 0.66 (95% CI, 0.55 to 0.80; p < 0.001) compared to those with high body weight fluctuation (Supplementary Figure 1).”
DISCUSSION:
While the study suggests that low BWF is associated with lower MetS risk, it cannot establish causality due to its observational nature..
Like your comment, we agreed that this study does not provide cause-and-effect relationship. These points were presented in the discussion section as follow line 359-359.
“Third, since this study is observational, there is no clear cause-and-effect relationship, and caution is needed in interpreting the findings.”
The study claims that "protein-rich foods might modulate the risk of metabolic syndrome incidents according to weight variability," but the data only show an association, not causation. The language used suggests a stronger causal relationship than the study can support.
We have thoroughly revised the entire Discussion section in response to the concerns you raised as follows, line 284-292, and line 335-347.
“This study identified differences in the consumption of protein-rich foods according to body weight fluctuations and suggested the possibility that protein-rich foods might modulate the incidence of metabolic syndrome according to body-weight variability, which is a novel strategy to prevent and manage metabolic syndrome in middle-aged Korean women. These meaningful results suggest that the incidence of metabolic syndrome is greater in the elderly population in Korea than in the middle-aged population, especially in women, compared with men. Its importance is also emphasized when considering that Korean women have a relatively low intake of protein-rich foods compared with other races or countries.“
“In this study, we found that the association between BWF and the incidence of MetS was modulated by protein-rich food consumption. Although studies have recently reported an association between weight variability and MetS in terms of the consumption of dietary carbohydrates and refined grains, our results are the first to address the consumption of protein-rich food groups [35]. A low weight fluctuation had a protective effect on the risk of developing MetS among subjects who consumed more total protein food groups and animal-source food groups. The reasons for these results are unclear. The reason for these results is unclear, but protein consumption is considered to be due to individual differences in body composition changes and related metabolic changes resulting from these changes in weight during long-term follow-up. That is, consuming protein-rich foods, especially animal protein-rich foods, could reduce weight fluctuations, but because body composition and metabolic changes vary from person to person, protein-rich food intake does not have a positive effect on the risk of MetS.”
It also states that this is a "novel strategy" for managing metabolic syndrome without sufficiently acknowledging previous research on protein intake and weight management.
In response to the points you raised, the Introduction section has been revised based on a review of existing literature.
The study acknowledges individual differences in metabolism and body composition changes but does not explore how these differences might impact the findings.
As this study is observational, it is difficult to fully examine individual differences in body composition changes. However, we believe that by focusing on middle-aged women, we have minimized the differences in body composition changes among individuals
The study reports that fish, shellfish, milk, and dairy products were associated with lower weight fluctuation, yet legumes and nuts did not show a similar effect. The discussion does not address why plant-based proteins might not contribute in the same way, leaving an important gap in the interpretation.
It has been modified based on the comment as follows, line 322-328.
“Animal protein, which contains high levels of essential amino acids, is usually considered to be more efficient at supporting muscle mass growth and recovery and maintaining muscle mass [33]. However, it is recommended to selectively supply protein foods because the intake of fish, milk and dairy products (low-fat foods) among animal-based protein and legumes and nuts among plant-based proteins lowers the risk of cardiovascular disease and is inversely correlated with hypertension, obesity, and insulin resistance [34].”
It is also unclear why certain animal-based proteins (e.g., meat and eggs) were not highlighted despite being included in the analysis
It has been modified based on the comment as follows, line 322-328, and 331-333.
“Animal protein, which contains high levels of essential amino acids, is usually considered to be more efficient at supporting muscle mass growth and recovery and maintaining muscle mass [33]. However, it is recommended to selectively supply protein foods because the intake of fish, milk and dairy products (low-fat foods) among animal-based protein and legumes and nuts among plant-based proteins lowers the risk of cardiovascular disease and is inversely correlated with hypertension, obesity, and insulin resistance [34].”
“In this study, subjects who consumed more fish, shellfish, milk or dairy products presented smaller fluctuations in body weight. These results suggest that protein sources such as fish, shellfish, or milk and dairy products may have the potential to maintain lean mass and result in a lower BWF.”
The study acknowledges some limitations, such as menopause-related weight changes and potential selection errors in protein-rich food groups. However, it does not sufficiently discuss:
Limitations were further presented.
The reliability of dietary assessments (e.g., self-reported food intake in FFQs).
It has been modified based on the comment as follows, as follows, line 161-165.
“Dietary intake was assessed using a semi-quantitative food frequency questionnaire (FFQ) developed and validated for the KoGES [23, 24]. The FFQ, which includes 103 food items, evaluated the frequency and portion size of food consumed over the past year. The correlation coefficients between the FFQ and a 12-day dietary record for various nutrients were approximately 0.45, indicating acceptable validity [24]. “
Reverse causation (i.e., whether individuals with stable weight patterns already have healthier dietary habits).
Generally, healthy dietary habits contribute to maintaining stable weight patterns, and after middle age, dietary habits tend to change less easily.
“
Unmeasured confounders (e.g., stress, exercise, or other dietary factors influencing weight fluctuations).
In the currently used KoGES data, data on stress factors are insufficient, and exercise was adjusted by calculating the usual activity level (including exercise). In addition, protein-rich foods were considered an important factor in weight change, so they were applied as independent variables in the study.
The study's inability to control for long-term adherence to dietary patterns—did participants consistently eat the same amount of protein over time?
The study was conducted with the assumption that changes in eating habits are uncommon after middle age
While the study suggests that consuming protein-rich foods may help reduce weight fluctuations, it does not provide clear guidelines on what amounts or types of protein should be consumed.
Since this study is observational, it does not provide sufficient data to offer clear guidelines. However, we believe it holds value as foundational data for future research.
It also fails to acknowledge potential risks of high protein intake, such as kidney strain or associations with other metabolic disorders.
We adjusted for the presence of diseases, and there were very few cases of kidney or liver diseases in the study population
The study does not provide clear recommendations on how findings can be applied in clinical or public health settings.
This study provides foundational data suggesting that protein-rich foods may help adjust the risk of metabolic syndrome associated with weight fluctuations

Round 2
Reviewer 2 Report
Comments and Suggestions for Authors
The authors have adequately responded and significantly improved the manuscript. The overall quality and clarity of the paper have greatly improved.